# ELASTICTOK: ADAPTIVE TOKENIZATION FOR IMAGE AND VIDEO

**Wilson Yan**[*,§] **Vlad Mnih**[†] **Aleksandra Faust**[†] **Matei Zaharia**[§] **Pieter Abbeel**[§] **Hao Liu**[*,†,‡]

[§]UC Berkeley  [†]Google DeepMind  [‡]Carnegie Mellon

## ABSTRACT

Efficient video tokenization remains a key bottleneck in learning general purpose vision models that are capable of processing long video sequences. Prevailing approaches are restricted to encoding videos to a fixed number of tokens, where too few tokens will result in overly lossy encodings, and too many tokens will result in prohibitively long sequence lengths. In this work, we introduce ElasticTok, a method that conditions on prior frames to adaptively encode a frame into a variable number of tokens. To enable this in a computationally scalable way, we propose a masking technique that drops a random number of tokens at the end of each frames's token encoding. During inference, ElasticTok can dynamically allocate tokens when needed – more complex data can leverage more tokens, while simpler data only needs a few tokens. Our empirical evaluations on images and video demonstrate the effectiveness of our approach in efficient token usage, paving the way for future development of more powerful multimodal models, world models, and agents. Video examples of using ElasticTok can be found on our website: largeworldmodel.github.io/elastictok

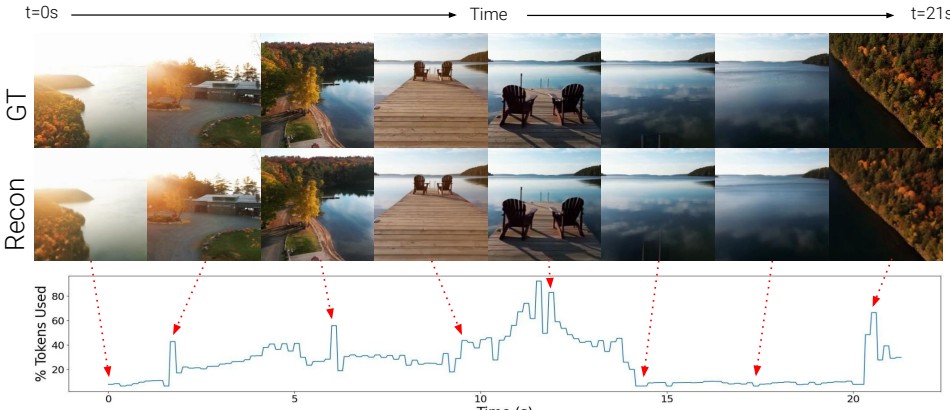

**Figure 1** **ElasticTok adaptively represent video based on information available**. (Top) Ground-truth video frames. (Middle) Reconstructed frames with varying token usage. (Bottom) The bottom section depicts how ElasticTok dynamically adjusts token allocation over time, with the percentage of tokens used correlating to different content complexities in the video.

## 1 INTRODUCTION

Efficient video tokenization remains a key bottleneck in learning general purpose vision models that are capable of processing long video sequences – a crucial aspect towards developing intelligent agents for the visual world. Prevailing approaches (Van Den Oord et al., 2017; Esser et al., 2021; Yu et al., 2023a; Yan et al., 2021) are restricted to encoding videos to a fixed number of tokens irrespective of the visual content of the original video. As a result, being able to reliably encode with little information loss requires increasing the number of tokens to account for the worst case, highly

---

[*] To whom correspondence should be addressed.

complex visual inputs. However, this in turn causes many wasted tokens on simpler data, and leads to significant computational challenges for downstream training, where encoding a video or trajectory can require tens or even hundreds of millions of tokens, resulting in substantial computational costs and inefficiency (Liu et al., 2024a; Brooks et al., 2024; Liu et al., 2024b). On the other hand, using a too few number of tokens will result in lossy encodings, and fundamentally limit the capabilities of vision models when processing more complex visual data.

Intuitively, we would want do learn a model that adaptively encodes visual input in a data-dependent manner to variable length encodings, similar to how existing works in image and video compression (Richardson, 2004; Li et al., 2023a; Christopoulos et al., 2000; Chen et al., 2017) compress data to a varying number of bytes. Taking inspiration from this, we introduce ElasticTok, a method that conditions on prior frames to adaptively encode a frame into a variable number of tokens. To enable this in a computationally scalable way, we propose a masking technique that drops a random number of tokens at the end of each frame's token encoding. During inference, ElasticTok can dynamically allocate tokens when needed – more complex data can leverage more tokens, while simpler data only needs a few tokens.

Our empirical evaluations demonstrate the effectiveness of ElasticTok, highlighted below.

- We show that ElasticTok can leverage adaptive tokenization to efficiently represent long videos with up to 2-5x fewer tokens.
- We show that ElasticTok enables flexibility in downstream vision-language tasks, allowing users to allocate tokens based on their compute budget.
- We show that ElasticTok allows leveraging different objectives during inference to adaptively trade off various semantic aspects of images.

## 2 BACKGROUND

### 2.1 BLOCKWISE RINGATTENTION

Blockwise RingAttention (Liu et al., 2024b) is a training technique for efficiently scaling transformers on long-context data. Each rank along the sequence parallel sharding axis retains non-overlapping, equal sizes slices of the full sequence. During an attention operation, each sequence parallel rank calculates its own corresponding $Q_i, K_i, V_i$, and computes its own portion of the full attention as: $\text{softmax}(Q_i^T[K_1, \ldots, K_S])[V_1, \ldots, V_S]$, where $S$ is size of the sequence parallel dimension. Instead

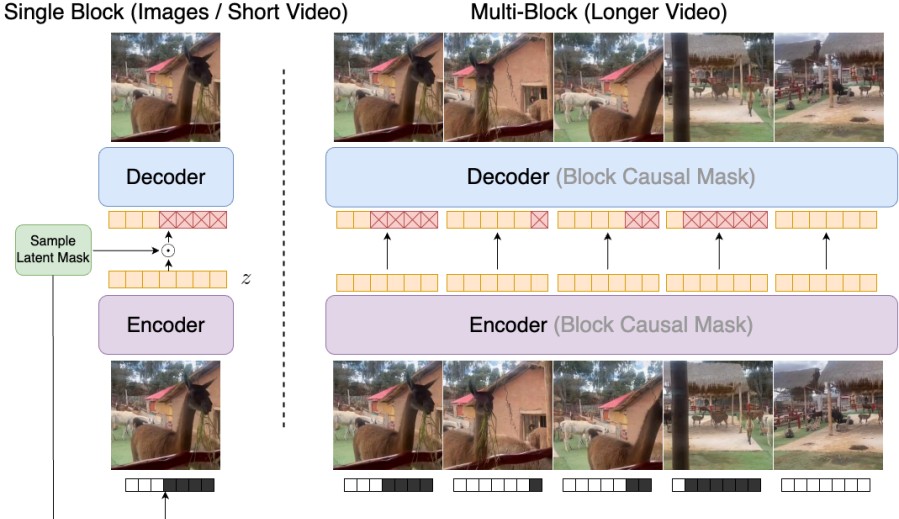

**Figure 2  ElasticTok adaptively encodes image and video to variable length outputs based on the complexity of the input data**. Single block uses an Encoder-Decoder pipeline with a sampled latent mask. Multi-block extends this with a Block Causal Mask to handle longer video sequences.

of naively all-gathering $K, V$, RingAttention proposes to iteratively pass $K_i, V_i$ blocks in a ring-structure, where rank $i$ will pass its current key-value block to rank $(i + 1) \mod S$. Importantly, RingAttention enables overlap between compute and communication by computing the partial attention over the current block while communicating the next block. In this paper, we leverage RingAttention to scale our method to long sequences of videos.

## 3 METHOD

In this section, we present ElasticTok, a scalable adaptive visual tokenizer than can efficiently encode image and video. At a high level, we leverage a specialized random masking scheme when training any standard autoencoder to learn our elastic encodings. In the following sections, we provide a more detailed description of our model – first covering the simpler single block case (image / short video), and then moving on to the multi block case (longer video). An overview of our method is shown in Figure 2.

### 3.1 TRAINING

First, we consider a single-block case in which we want to learn elastic codes over an image or a short video chunk (*e.g.*, 4 frames). Let $x$ represent a sequence of input frames, where images are considered as 1 frame videos. Our proposed method extends upon any existing variants of autoencoders (VQ-VAE (Van Den Oord et al., 2017), FSQ (Mentzer et al., 2024), VAE (Kingma, 2013)) by incorporating additional masking of the encoder tokens.

For each training input $x$, we first uniformly sample the number of tokens to keep $\ell \sim U(\{M_{min}, \ldots, M_{max}\})$, where $M_{min}$ and $M_{max}$ define the supported range of encoding lengths. $M_{max}$ is generally set as $N$, the maximum number of tokens the encoder can output (*e.g.*, 2048 or 4096), and $M_{min}$ a lower bound such as 128 or 256. We found this lower bound necessary as we found that sampling too few tokens could destabilize training. After sampling $\ell$, we then compute its mask $m$, defined as a binary vector of length $N$ with the first $\ell$ values set to 1. Input $x$ and mask $m$ are then fed into the encoder $E(x, m)$ to compute $z$. $z$ is masked as $z_m = z \odot m$, and fed through the decoder $D(z_m)$ to produce $\hat{x}$. Finally, our model optimizes a reconstruction loss, with potentially auxiliary losses depending on the exact autoencoder being used (*e.g.*, KL for VAE or VQ loss for VQ-VAE). Although our method is generally architecture invariant, we opt to use ViTs (Dosovitskiy, 2020) as our encoder and decoders for simplicity.

We can extend our method to process longer video by breaking up video into blocks, with sizes defined by the number of frames used for the single block model (e.g 4 frames per block). The training process remains the same as the single block case, with two key differences: (1) masks for each multi block video sequence are sampled independently for each block, and (2) a block causal mask (block size $N$) is used in the encoder and decoder. The number of blocks is constrained only by the transformer's context size. We utilize a prior long-context method (Liu et al., 2024b) to train ElasticTok on longer videos. Architecturally, there are no added parameters which we found useful when progressively training our model from single block to multi block.

Exact details of our model's forward pass are described in Algorithm 1.

### 3.2 INFERENCE

There are two primary ways to use ElasticTok for inference – by specify a target encoding length, or a target reconstruction threshold.

**Target encoding lengths**. This method is simple to implement, as it only requires generating the correct mask and running the input through the encoder. However, although such inference is simple and efficient, it is difficult from a user's perspective due to lack of knowledge of how many tokens to specify.

**Target reconstruction threshold**. This method presents a more intuitive inference approach that will automatically adaptively allocate tokens between different inputs based on a specified reconstruction threshold (*e.g.*, target pixel-wise MSE loss). Visual content that is easy to reconstruct may require fewer tokens to meet the threshold whereas more complex inputs may require more tokens. Doing so

requires a search process over different encoding lengths to determine lowest encoding length that still satisfies the given threshold.

We consider a few different search algorithms, detailed below:

- **Full Search**: Exhaustive search over every possible number of tokens lengths, and treat the result as ground truth. We use our discrete latent with max 4096 tokens.
- **Binned Search (100)**: Similar to Full Search, but only evaluate on 100 uniformly spaced token lengths.
- **Binary Search**: We perform search using binary search, where the token length is the "array index", and the evaluated reconstruction loss is the "array value." Note that this assumes that reconstruction loss is monotonic with respect to token length, which is generally close to true, but not fully accurate, hence error in the result produced.
- **Neural Regression**: We collect 100k examples of paired (image / video, token length) data, and finetune our autoencoder to directly regress the number of tokens given an image or video.

Each search algorithm has its own trade-off between accuracy and efficiency – we further examine this relationship in Section 4.4.

In the multi block case, we iteratively perform the search process for each block in a block autoregressive manner, using caching similar to in language models. Algorithm 2 provides more details on the exact inference process.

Further ablations (Table 5 in Appendix F) suggest that we can additionally achieve 2x inference speedup at a slight cost to encoding quality by not conditioning the mask on the encoder – this would require only running the encoder once, and then performing iterative search through masking the encoding and only running the decoder. This may be a better choice depending on the size of the downstream pretrained model.

---

**Algorithm 1** Training

**Required:** Video $x$. Patch size $T_p \times H_p \times W_p$.
**Required:** Tokens per block $Z$
**Required:** Encoder $E$. Decoder $D$
**Required:** Min / Max token lengths $M_{min}, M_{max}$
// batch size, timesteps, height, width, channels
// $x$ is $B \times T \times H \times W \times C$
$x \leftarrow \text{PatchifyRearrange}(x)$ // $B \times L \times D$
// $L = T/T_p * H/H_p * W/W_p$
// $D = T_p * H_p * W_p * C$
$N_b \leftarrow L/Z$
**for** $i$ in $\{0, \ldots, N_b - 1\}$ **do**
    Sample $\ell_i \sim U(\{M_{min}, \ldots, M_{max}\})$
    Initialize masks $m_i \leftarrow \mathbf{0}_Z$
    Fill masks $m_i[: \ell_i] = 1$
**end for**
Concatenate masks $m \leftarrow (m_0, \ldots, m_{N_b-1})$
Encode $z \leftarrow E(x, m)$ // $B \times L \times D_z$
Mask out along sequence length $z_m \leftarrow z \odot m$
Decode $\hat{x} \leftarrow D(z_m)$
Compute loss $\mathcal{L}(\hat{x}, x)$ // *e.g.*, MSE + LPIPS

---

**Algorithm 2** Inference

**Required:** Video $x$. Patch size $T_p \times H_p \times W_p$.
**Required:** Tokens per block $Z$.
**Required:** Encoder $E$. Decoder $D$
**Required:** Target reconstruction threshold $t$
// batch size, timesteps, height, width, channels
// $x$ is $B \times T \times H \times W \times C$
$x \leftarrow \text{PatchifyRearrange}(x)$ // $B \times L \times D$
// $L = T/T_p * H/H_p * W/W_p$
// $D = T_p * H_p * W_p * C$
$N_b \leftarrow L/Z$
Initialize KV cache $c$ of length $L$
**for** $i$ in $\{0, \ldots, N_b - 1\}$ **do**
    // See Section 2.2 for search algorithms
    $\ell_i, m_i, c \leftarrow \text{SearchAlgo}(x_{iZ:(i+1)Z}, c, t, D, E)$
    Encode $z_i \leftarrow E(x_{iZ:(i+1)Z}, m_i, c) \odot m_i$
**end for**
Return $z \leftarrow (z_0, \ldots, z_{N_b-1})$

---

# 4 EXPERIMENTS

In this section, we introduce our evaluation setup and present the results of pretraining ElasticTok to adaptively represent images and videos, as well as its performance on downstream tasks.

## 4.1 EXPERIMENTAL SETUP

**Model details**. We train 200M parameter discrete and continuous autoencoders on images and video sequences. For learning discrete representations, we use FSQ (Mentzer et al., 2023), while

for continuous representations, we use a VAE (Kingma, 2013). Both models employ ViT encoders and decoders, consisting of a single patchify strided convolutions (the transpose for the decoder), followed by transformer blocks identical to those in LLaMA 2 (Touvron et al., 2023). Attention is applied in a block-causal manner, with block size equal to the number of tokens per video block. The encoder is additionally conditioned on the token mask (represented as a binary mask) by replacing 0's and 1's with different learned embedding vectors, and then added to the video patchify output hidden states. The bottleneck representations are computed through a linear projection on the encoder output. Similar to prior works (Esser et al., 2021), we optimize both a pixel-wise reconstruction loss (MSE) and a perceptual loss (LPIPS) (Johnson et al., 2016). Notably, we do not use a GAN (Goodfellow et al., 2014), as we find it more difficult to stably train when progressively extending to much longer sequences (e.g. 1000+ frames), which we leave to future work, as it is not the main focus of this paper.

**Training details**. All models are jointly trained on $256 \times 256$ images and 24 FPS videos. Each video block consists of 4 frames, and joint training is conducted by treating images as single-frame videos, replicating each image 4 times to match the block size. Each block is encoded into 2048 or 4096 tokens by the continuous and discrete autoencoders, respectively. Following prior research in LWM (Liu et al., 2024a) on scaling sequence length, we begin by training our models on single-block cases (images and short videos) and progressively extend the context length to cover up to 40-second videos (512K to 1M tokens). We train our long video models using v4-512 TPUs from Google Cloud on the COYO-700M image dataset and a custom dataset consisting of 6M videos scraped from the web. We additionally train an image-only model on ImageNet using v4-256 TPUs. Details for further training details can be find in Appendix B, and data details in Appendix C

**Baselines**. We compare our proposed method against fixed token baselines trained at a varying token capacities. The architecture, training processes, and total FLOPs are exactly the same as our method with the only difference being restricted to one fixed mask, instead of sampling variable masks.

**Evaluation Search Algorithm** Unless otherwise noted, all results use the Binary Search algorithm as described in Section 3.2.

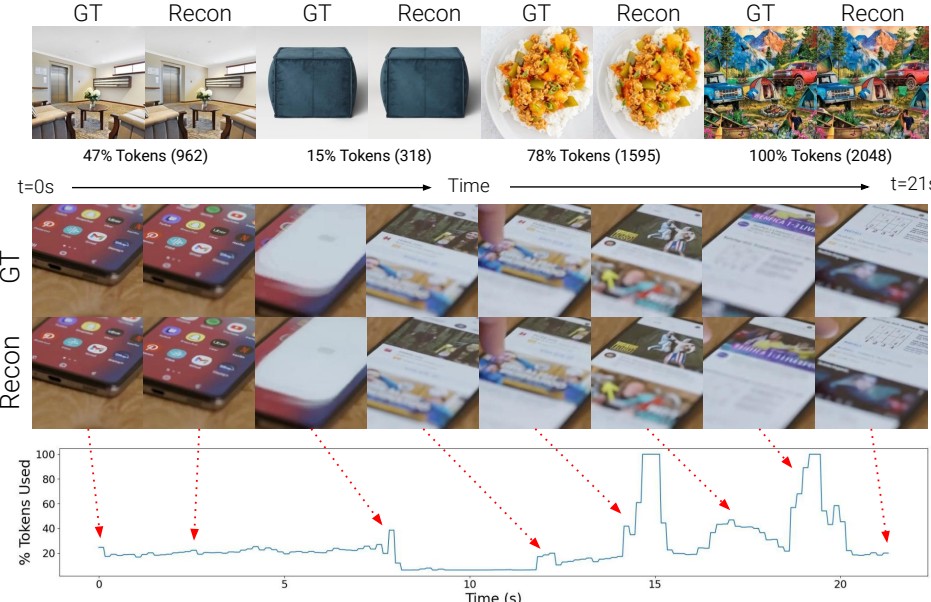

**Figure 3  ElasticTok adaptively encodes image and video to variable length outputs based on the complexity of the input data (using ElasticTok-VAE)**. The top rows shows examples of ElasticTok on images. Below shows a video example with: (Top) Ground-truth video frames. (Middle) Reconstructed frames with varying token usage. (Bottom) The bottom section depicts how ElasticTok dynamically adjusts token allocation over time, with the percentage of tokens used correlating to different content complexities in the video.

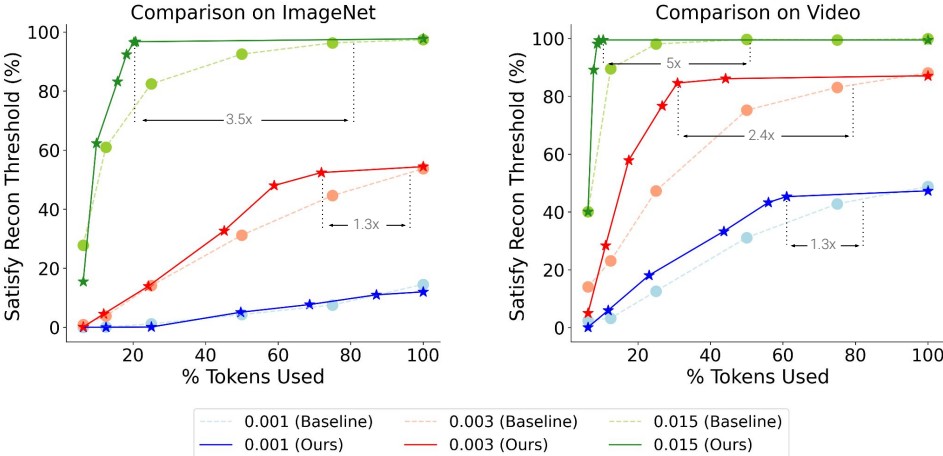

**Figure 4    Performance comparison between baseline and ElasticTok-FSQ on ImageNet and Video.** The y-axis shows the percentage of samples that satisfy the reconstruction threshold, while the x-axis represents the percentage of tokens used. (Left) On image, ElasticTok achieves a 3.5x and 1.3x efficiency boost at different reconstruction thresholds. (Right) On video, ElasticTok shows a 5x and 2.4x improvement over the baseline, maintaining superior performance while using fewer tokens. Figure 10 in Appendix D shows reference examples of reconstruction quality for an image at different thresholds.

## 4.2    MAIN RESULTS

Intuitively, the advantages of our elastic tokenization allow variable length codes that can depend on the visual content of a given image or video. Figure 3 shows some qualitative examples that demonstrate this. All images and videos are provided the same MSE reconstruction quality (0.003) threshold to satisfy, and require different number of tokens depending on the complexity the images. Simpler images such as the blue cushion require fewer tokens, while more complex images such as the painting or recipe have details that require more tokens to properly encode. For the provided video example, we generally see larger token usage spikes in the event of a scene change or faster motion, such as when the phone animates a transition screen, or when the finger scrolls up in the newsfeed. More qualitative examples can be found at: largeworldmodel.github.io/elastictok.

Figure 4 shows quantitative comparisons between our method and different fixed token baselines trained at each token length. In order to show the benefits of elastic token representations, we compute a quantitative metric that measures the percentage of images or videos blocks in which a model satisfies a specified reconstruction (MSE) threshold. As shown, our method can leverage its elastic representations to achieve similar reconstruction satisfactory percentages compared to baselines while using 1.3x - 5x fewer tokens, depending on the given threshold. Generally, more lax (0.015) reconstruction thresholds present larger performance improvements due to more room to use fewer than max tokens. In contrast, more strict (0.001) thresholds usually almost always require all tokens. Different thresholds may be useful for different cases, where more lossy encodings can be used for simple VQA tasks, while more accurate reconstructions are useful for tasks such as image / video generation. Table 3 shows similar results for the continuous variant of our model, compared against a single baseline model fixed to $50\%$ token usage.

Figure 5 shows qualitative examples of progressive reconstruction quality as we increase the number of tokens. Different visual inputs will saturate reconstruction quality at a different number of tokens. This flexibility allows us to use a large number of tokens for very hard images (*e.g.*inputs with detailed fine-grained text), while saving tokens on inputs that are easier to reconstruct. Figure 6 shows the performance of our method as we increase the sequence length of the model – performing better as sequence length increases due to being able to leverage long context for more accurate reconstruction. However, performance degrades at the max context length (1M tokens), which we hypothesize is due to not enough training due to our relatively limited compute budget for that context size.

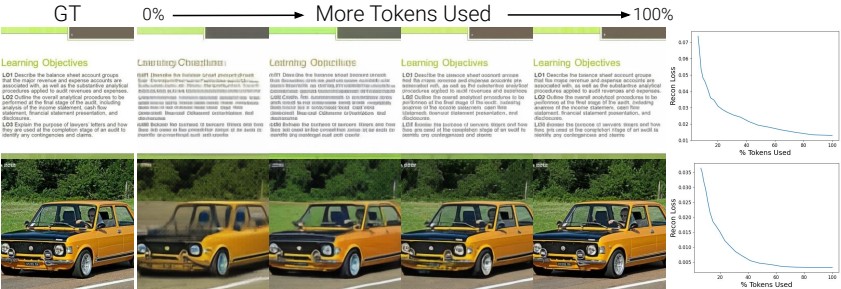

**Figure 5  Loss progressively declines as more tokens are used (ElasticTok-FSQ).** The top row illustrates the impact on text clarity, while the bottom row shows the effect on image sharpness. The graphs on the right quantify the reconstruction loss relative to token usage percentage, showing a rapid decline as more tokens are consumed.

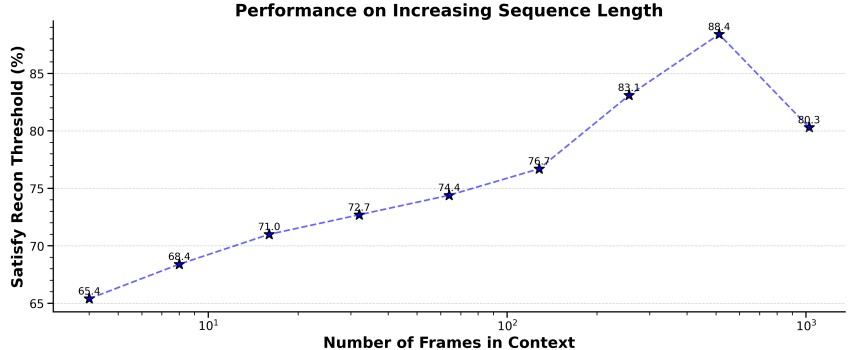

**Figure 6  Progressive performance increase with more frames (ElasticTok-FSQ).** Performance improves with increasing sequence length, peaking around 100 frames before a slight decline. (Note the log scale for the x-axis)

> **Takeaway:** ElasticTok offers significant advantages in image and video reconstruction by using variable-length codes that depend on content complexity. Qualitatively, simpler images require fewer tokens, while more complex ones need more tokens. Quantitatively, ElasticTok achieves similar reconstruction quality compared to fixed token baselines while using up to 5x fewer tokens

## 4.3  DOWNSTREAM TASKS

In order to evaluate the quality of our learned representations, we finetune a pretrained language model with our visual tokens, and evaluate on text-image and text-video VQA tasks. We use OpenLLaMA-3B (Geng and Liu, 2023), and finetune with visual tokens from the continuous variant of our model (max 2048 tokens). We finetune the entire model for 80M text-image pairs from COYO-700m (Byeon et al., 2022), and chat finetune with data from Chen et al. (2023). For video, we continue to train on WebVid10M (Bain et al., 2021) and finetune on Video-ChatGPT (Maaz et al., 2023) instruct data. Figure 7 shows evaluation results on GQA (Ainslie et al., 2023), POPE (Li et al., 2023b), MSVD-QA (Xu et al., 2017), and MSRVTT-QA (Xu et al., 2017) at a varying number of input inference tokens. Benchmark performance generally increases at the number inference tokens increases, suggesting the usefulness of our model as a means for users to potentially be able to flexibly choose how many tokens use, as a compute / accuracy trade-off, especially useful for users with more limited compute / API call budgets. Lastly, Table 1 shows that our VLM finetuned on our adaptive tokens can match the performance of a VLM finetuned on a fixed token (50% tokens) baseline tokenizer, suggesting that there is little to no loss in accuracy when switching to ElasticTok as the tokenizer, with additional added adaptivity benefits.

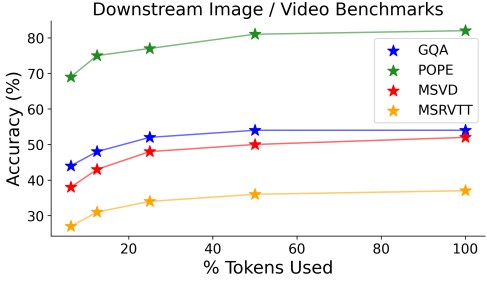

**Figure 7  The accuracy and compute trade-off with varying percentages of tokens used (ElasticTok-VAE).** This allows users to adjust the accuracy based on computational budget.

|          | GQA | POPE | MSVD | MSRVTT |
|----------|-----|------|------|--------|
| **Ours**     | 54% | 82%  | 52%  | 37%    |
| **Baseline** | 54% | 82%  | 53%  | 37%    |

**Table 1  Comparison of our method with baseline on image and video benchmarks (ElasticTok-VAE).** Our method can match the performance of the baseline trained on a fixed number (100%) of tokens. However, baseline models are restricted to a fixed token output, and require full pretraining a new model for every possible token length, whereas ElasticTok only requires a single model to generalize to all token lengths.

## 4.4 INFERENCE

**Speed**. Table 2 shows comparisons of the accuracy and inference speed (NFEs) for each of these methods. Error rate is computed as the relative error between the ground truth number of tokens used, and the number of tokens returned by each method. NFE (Number of Function Evaluations) is the number of forward passes that are needed. In generally, methods closer to exhaustive search (Full / Binned Search) are more accurate, while methods that orders of magnitudes faster generally have much higher error around $5 - 10\%$. Users with less compute available for inference may want to use the faster inference methods at a slight cost to token encoding accuracy. Users with a lot of compute can leverage more exhaustive approaches while retaining fast inference speed by computing search for token lengths as parallel batches.

**Target Objective**. One benefit of computing elastic tokens is that we can switch to any target objective during inference, and can adaptively tokenize visual content based on certain visual preferences, or aspects that users want to prioritize by allocating more tokens to. For example, Figure 8 shows qualitative examples comparing running inference using an MSE objective versus using a CLIP loss (image-image cosine distance) On average, thresholds are set so that both models as the same average number of tokens over the dataset, but OCR capabilities of CLIP allow it to show preference towards allocating more tokens in reconstructing the text (bottom images), while deprioritizing other images such as the fine-grained details in the bottom image. In general, any scalar function is usable, and does not need to be differentiable (*e.g.*, running OCR detection, and computing a more direct text reconstruction metric, or running segmentation and priortizing object clarity).

> **Takeaway:** ElasticTok allows multiple adaptive inference methods: full search is the most accurate but slow, while faster methods like neural regression slightly sacrifice accuracy (5-10% error) for speed. Users with less compute can opt for faster methods with a slight accuracy loss. Additionally, inference objectives can be adapted to prioritize specific content, such as focusing on text over other image features, allowing for flexible token allocation based on user preferences.

## 4.5 FREQUENCY ANALYSIS

We hypothesize that visual content that tends to have more fine-grained details, and high-frequency features tends to use a higher number of tokens. To investigate this, we run inference on 2k videos (5s, 32 blocks), and also compute an approximate frequency metric for each video block. For each frame in a video block, we convert it to grayscale, run a Sobel edge detection filter (horizontal / vertical), and compute average gradient magnitudes. Figure 9 shows scatter plots and correlation for the single and multi block cases. For the single block case, the correlation between encoding length and amount of high-frequency detail is quite high (0.77). The multi block case is lower (0.67), which is most likely

due to slight decorrelation from being able to leverage past frames (conditional encoding) in videos.

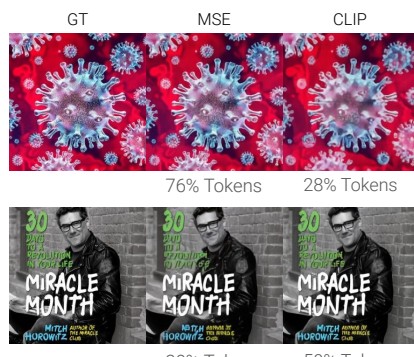

| Inference Method | Error Rate | NFEs |
|---|---|---|
| Full Search | $0\%$ | 4096 |
| Binned Search (100) | $0.5\%$ | 100 |
| Binary Search | $7\%$ | 12 |
| Neural Regression | $9\%$ | 1 |

**Table 2   Comparison of inference methods showing their respective error rates, number of function evaluations (NFEs) (ElasticTok-FSQ**. Note that while Full and Binned Search are more computationally expensive, they could also benefit more from parallel function evaluations if compute is available.

**Figure 8   Comparison of different loss functions for inference (ElasticTok-FSQ)**. Inference with a CLIP model (cosine distance) prioritizes textual reconstruction (bottom) while deprioritizing other detailed visual features (top).

> **Takeaway:** Pearson correlation shows that ElasticTok adaptively allocate number of tokens for videos with more detailed visual content and high-frequency features require more tokens for encoding.

## 5   RELATED WORK

**Adaptive Representation**. There have been a lot of research studies on learning adaptive or ordered representations. Early work on nested dropout (Rippel et al., 2014) proposes using dropout (Srivastava et al., 2014) in the context of autoencoders to learn ordered representations. In this approach, an index is sampled from a prior distribution, and all units with an index larger than the sampled one are dropped. Similar to our ElasticTok, this method imposes an inherent ordering on the representation dimensions. Units that are dropped less frequently encode more important information, while those that are dropped more often encode less critical details. Other works study adaptive architectures based on context size (Kim and Cho, 2020), slimmable neural networks that train a network by sampling multiple sub-networks of different channel numbers simultaneously, where the weights are shared among different widths (Lee and Shin, 2018; Yu et al., 2018), adaptive width and depth in BERT (Hou et al., 2020), or dropping random layers during training to increase robustness to pruning (Fan et al., 2019; Huang et al., 2016). Dieleman et al. (2021) learns a variable-rate representation on audio applied to speech. Transframer (Nash et al., 2022) uses variable-length DCT representations with transformers for general image and frame-level video prediction tasks. Matryoshka representation learning (Kusupati et al., 2022) explores adding nested substructures inside the standard Transformer block. Graves (2016) explores using recurrent neural networks to learn adaptive computation for learning tasks. However, none of the previous works consider learning elastic representations in autoencoders for long sequences like videos. Our work provides a scalable solution for representing videos with elastic representations.

**Tokenization**. Representing visual images with a set of discrete tokens has been extensively studied, such as in VQVAE (Van Den Oord et al., 2017; Razavi et al., 2019) and VQGAN (Yu et al., 2021). Recent research has highlighted several shortcomings of traditional tokenization methods, including low vocabulary utilization. In response, alternative approaches like FSQ have been explored (Mentzer et al., 2024; Yu et al., 2023b). These discrete visual tokens facilitate integration with next-token prediction in language models and multimodal models (Yu et al., 2023b; Liu et al., 2024a; Xie et al., 2024; Wang et al., 2024). Parallel to this, other studies have investigated the use of continuous embeddings for representing images and videos (Lin et al., 2023; Liu et al., 2023; Zhu et al., 2023). Other research proposes next-scale prediction (Tian et al., 2024), which employs hierarchical multi-scale modeling to first capture coarse details, followed by finer ones. However, this approach requires a manually defined hierarchy and uses a fixed number of tokens. Our work, while complementary

to these efforts, focuses on adaptive representation and can be directly applied to both discrete and continuous approaches.

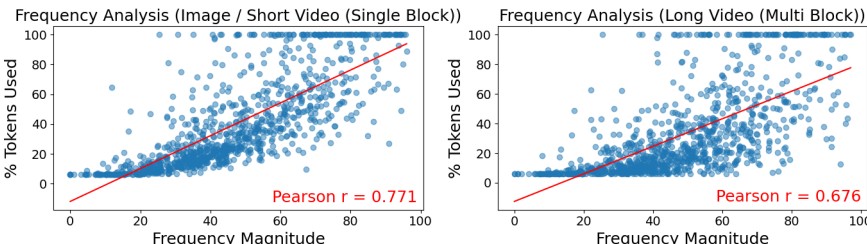

**Figure 9** **Comparison of token usage versus frequency magnitude in single-block and multi-block frequency analysis (ElasticTok-FSQ)**. Both scatter plots show a strong positive correlation between frequency magnitude and token usage in a single-block setting a multi-block setting. The red lines represent the linear regression fits for each case.

**Compression**. Adaptive learning for images and video have also been well-studied in the context of variable-rate compression. JPEG (Christopoulos et al., 2000) remains one of the most popular lossy image compression algorithms in the world, using a combination of DCT with quantization followed by entropy coding. For video, codecs such as H264 (Wiegand et al., 2003), H265 (Sze et al., 2014), VP9 (Mukherjee et al., 2015), and AV1 (Han et al., 2021) are popularly used compression algorithms. Prior works have additionally explored extending neural networks to learn more effective compressors. Theis et al. (2022) and Minnen et al. (2018) introduce methods that leverage CNNs to learn effective variable bit-rate compression algorithms over images competitive with JPEG. Li et al. (2023a), Mentzer et al. (2022), and Li et al. (2021) extend neural networks to learn how to effectively compress videos. While similar to our work in that these methods also learn adaptive encodings, these compression methods are more difficult to utilize for downstream training and generation tasks due to lack of direct compatibility. In contrast, our work builds upon existing tokenization strategies (VAE, FSQ) that have shown to work well for such downstream tasks, as well as take advantage of the adaptive representations to better utilize compute.

## 6 DISCUSSION AND CONCLUSION

In this work, we propose an elastic representation approach to address the inefficiencies of traditional video encoding approaches through an adaptive encoding method that selectively encodes new information based on the context of previous frames. By dynamically allocating resources, it reduces computational costs while maintaining high-quality video representation. The proposed technique of dropping tokens at the end of each sequence allows the model to prioritize essential information, ensuring scalability and efficiency during inference. ElasticTok demonstrates strong performance in both video representation and downstream tasks. Lastly, we identify some limitations of our method, as well as possible directions for future work.

- **Masking schemes**: Our model currently slightly underperforms baselines at the tail ends of encoding lengths, which we hypothesize may be partially due to conflicting representations that the model needs to learn in each case (low frequency, global encodings vs high frequency, local encodings). Preliminary investigations showed that changing the way tokens are distributed (as opposed to our method of left-aligned tokens) improved reconstruction accuracy. Future work may also explore learnable encoding masks to dynamically select which tokens to keep for each input.

- **Other temporal modalities**: Although our primary focus was on image and video, our method is generally modality-agnostic and can be extended to any temporal data, such as audio and decision-making trajectories (e.g., state, action, reward).

- **Conditional encoding and different training objectives**: Our focus was on leveraging temporal encoding to achieve more efficient tokenization while retaining high reconstructed visual quality. However, our method can be explored more broadly as a means of encouraging meaningful lossy encodings. For example, robotics models can learn text-conditioned visual encoders that capture only task-relevant information from an image. In other scenarios, different reconstruction objectives could prioritize facial or textual reconstruction while ignoring background reconstruction accuracy.

## ACKNOWLEDGMENTS

We gratefully acknowledge the Google TPU Research Cloud (TRC) program for providing access to TPUs. Pieter Abbeel holds concurrent appointments as a Professor at UC Berkeley and as an Amazon Scholar. This paper describes work performed at UC Berkeley and is not associated with Amazon. We extend our thanks to Xinyang Geng, Jimmy Shi, Tim Rocktäschel, Joao Carreira, and Satinder Singh for their valuable discussions and insightful feedback on an earlier version of this paper, and to Doina Precup for valuable support of this work.

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

## A  MODEL CONFIGURATION

|  | ElasticTok-VAE (Continuous) | ElasticTok-FSQ (Discrete) |
|---|---|---|
| Parameters | 210M | 210M |
| Frame Resolution | $256 \times 256$ | $256 \times 256$ |
| Block Size ($M_{max}$) | 2048 tokens (4 frames) | 4096 tokens (4 frames) |
| $M_{min}$ | 128 tokens | 256 tokens |
| Max Number of Frames | 1024 | 1024 |
| Patch Size (T, H, W) | (2, 8, 8) | (1, 8, 8) |
| Hidden Size | 1024 | 1024 |
| FFN Size | 2048 | 2048 |
| Encoder Layers | 10 | 10 |
| Decoder Layers | 10 | 10 |
| RoPE Theta | 5000000 | 50000000 |
| Max Sequence Length | 512K | 1M |
| FSQ Dims | N/A | 8, 8, 8, 5, 5, 5 (64k codes) |
| VAE Dim | 8 | N/A |
| KL Weight | 1e-8 | N/A |

## B  TRAINING DETAILS

The tables below showing trainin details for each of our models. Our Long Video model is trained on a mix of images and video (Batch Split), and each run (e.g. Long Video (2)) is initialized from the previous run (e.g. Long Video (1)). For the discrete (FSQ) model, each block has 4k tokens (256 blocks = 1M tokens), and the continuous (VAE) model has 2k tokens in each block (256 blocks = 512K tokens).

|  | ImageNet | Long Video (1) | Long Video (2) | Long Video (3) |
|---|---|---|---|---|
| Batch Size | 256 | 256 | 256 | 256 |
| # Blocks | 1 | 1 | 2 | 4 |
| Batch Split (Image / Video) | 100%/0% | 50%/50% | 50%/50% | 25%/75% |
| Total Iterations | 200k | 200k | 80k | 50k |
| Learning Rate | $2 \times 10^{-4}$ | $2 \times 10^{-4}$ | $2 \times 10^{-4}$ | $2 \times 10^{-4}$ |
| Optimizer | AdamW | AdamW | AdamW | AdamW |
| Weight Decay | $1 \times 10^{-4}$ | $1 \times 10^{-4}$ | $1 \times 10^{-4}$ | $1 \times 10^{-4}$ |
| Warmup Iterations | 10k | 10k | 5k | 2k |

|  | Long Video (4) | Long Video (5) | Long Video (6) | Long Video (7) |
|---|---|---|---|---|
| Batch Size | 64 | 16 | 8 | 4 |
| # Blocks | 16 | 64 | 128 | 256 |
| Batch Split (Image / Video) | 25%/75% | 25%/75% | 25%/75% | 25%/75% |
| Total Iterations | 10k | 1k | 500 | 200 |
| Learning Rate | $2 \times 10^{-4}$ | $2 \times 10^{-4}$ | $2 \times 10^{-4}$ | $2 \times 10^{-4}$ |
| Optimizer | AdamW | AdamW | AdamW | AdamW |
| Weight Decay | $1 \times 10^{-4}$ | $1 \times 10^{-4}$ | $1 \times 10^{-4}$ | $1 \times 10^{-4}$ |
| Warmup Iterations | 1k | 200 | 100 | 25 |

## C  DATA CURATION DETAILS

**Image Data**  We use COYO-700M (Byeon et al., 2022) for our text-image data. We filter out images with less than $256 \times 256$ images. After accounting for stale links, we are left with roughly 350M

text-image pairs. For better text correspondance, we further generate synthetic captions for each image using the mooondream2 model.

**Video Data** To collect our video data, we first scrape Common Crawl to collect video links. After deduplication, we download each video and split then into individual scenes using PySceneDetect. For each scene we run various filtering metrics, such as OCR detection, NSFW scoring, motion score, and aesthetic scoring. Filtering metrics are aggregated over each scene in a video, and used to select a final subset of 6M videos ranging from 4 seconds to 2 minutes in length.

## D    RECONSTRUCTION EXAMPLES BY MSE

We include the reconstructed images at different Mean Squared Error (MSE) loss thresholds in Figure 10 as a reference for how the images appear at various MSE thresholds.

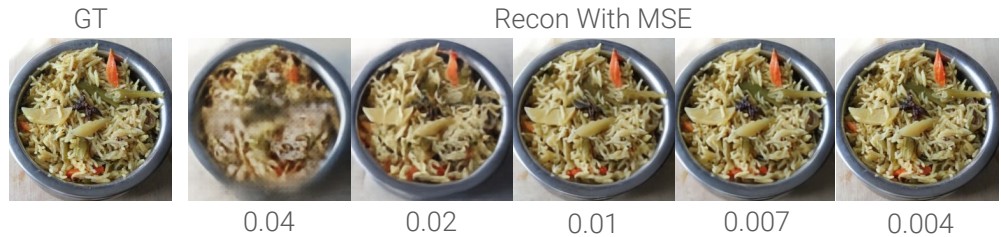

**Figure 10    Comparison of reconstructed image quality at varying MSE thresholds**. The ground truth (GT) image is displayed on the left, followed by reconstructions at different MSE thresholds (0.04, 0.02, 0.01, 0.007, and 0.004), showing progressive improvement in fidelity as the threshold decreases.

## E    DETAILS ON NEURAL REGRESSION MODEL

To collect training data, we first run inference with the tokenizer for a pre-defined noise target of 0.003 on 100K sequences to get pairs of image/video and token lengths. To train the neural regression model, we augment the pretrained encoder model with alternating downsampling and transformer blocks. We downsample by a factor of 4 over sequence each time each time using a simple size 4 stride 4 1D conv, and each set of transformer blocks consists of 4 layers. The final output is pooled, and fed through a 2 hidden layer MLP to output a single scalar per block. The regressor is optimized using MSE loss, and regresses to a normalized token length in the range $[-1, 1]$.

## F    FURTHER EXPERIMENTS

**Table 3    Performance of ElasticTok-VAE on videos**. Values in the table show the percentage of reconstructed video blocks that satisfy a given reconstruction threshold. The baseline is a 50% fixed token baseline, and our method uses variable token lengths with an average of 50% token usage over the dataset.

|  | Threshold | | |
|---|---|---|---|
|  | 0.001 | 0.003 | 0.015 |
| **Baseline** | 40% | 78% | 99% |
| **Ours** | 50% | 88% | 99% |

Table 3 above shows the same metrics as Fig 4 but on our continuous variant ElasticTok-VAE model. Similar to ElasticTok-FSQ, the VAE variant can reliably encode more videos to satisfy a given

reconstruction threshold while using the same number of tokens (on average) as a FLOPs identical continuous variant baseline.

**Table 4   Quality baseline comparison with ElasticTok-FSQ on videos**. Values shown in the table represent the average percentage of tokens used by each model to achieve the same worse-case MSE (compute MSE over each video block, and take the max MSE over the all blocks in the video). Our method uses fewer tokens to acheive the same worse-case reconstruction as the baseline.

|  | Threshold | | |
|---|---|---|---|
|  | **0.001** | **0.003** | **0.015** |
| **Baseline** | 78% | 57% | 15% |
| **Ours** | **75%** | **37%** | **9.4%** |

Table 4 above shows running a quality baseline comparison between our method and a baseline. For our method, we encode each evaluation video using a given reconstruction threshold, and compute the average percentage of tokens per frame. For the baseline, we find the needed fixed tokens per frame such that the worse-case reconstruction MSE over the whole video matches the worst-case MSE of ElasticTok's reconstruction. Intuitively, fixed token baselines perform worse due to allocated a fixed number of tokens per video block, and must optimize for the worst-case scenario. In contrast, ElasticTok can save tokens when the videos transition to simpler to encode scenes while allocating more tokens to more complex scenes.

**Table 5   Ablation comparing with and without encoder conditioning on the masked tokens**. Values in the table show the percentage of reconstructed video blocks that satisfy a given reconstruction threshold under the constraint that both models use the same average percentage of tokens.

|  | Threshold | | |
|---|---|---|---|
|  | **0.001** | **0.003** | **0.015** |
| **With Enc Cond** | **36%** | **79.6%** | **98.5%** |
| **Without Enc Cond** | 33.1% | 75.3% | 98.4% |

Table 5 shows a comparison of ElasticTok with and without encoder conditioning on the mask. ElasticTok with conditioning slightly outperforms the model without conditioning. As such, it may be more efficient to train the model without encoder conditioning depending on expected size of the downstream models - as this present a trade-off in terms of slightly better encoding quality at the cost of 2x slower inference speed.

## G  MORE PROGRESSIVE RECONSTRUCTION EXAMPLES

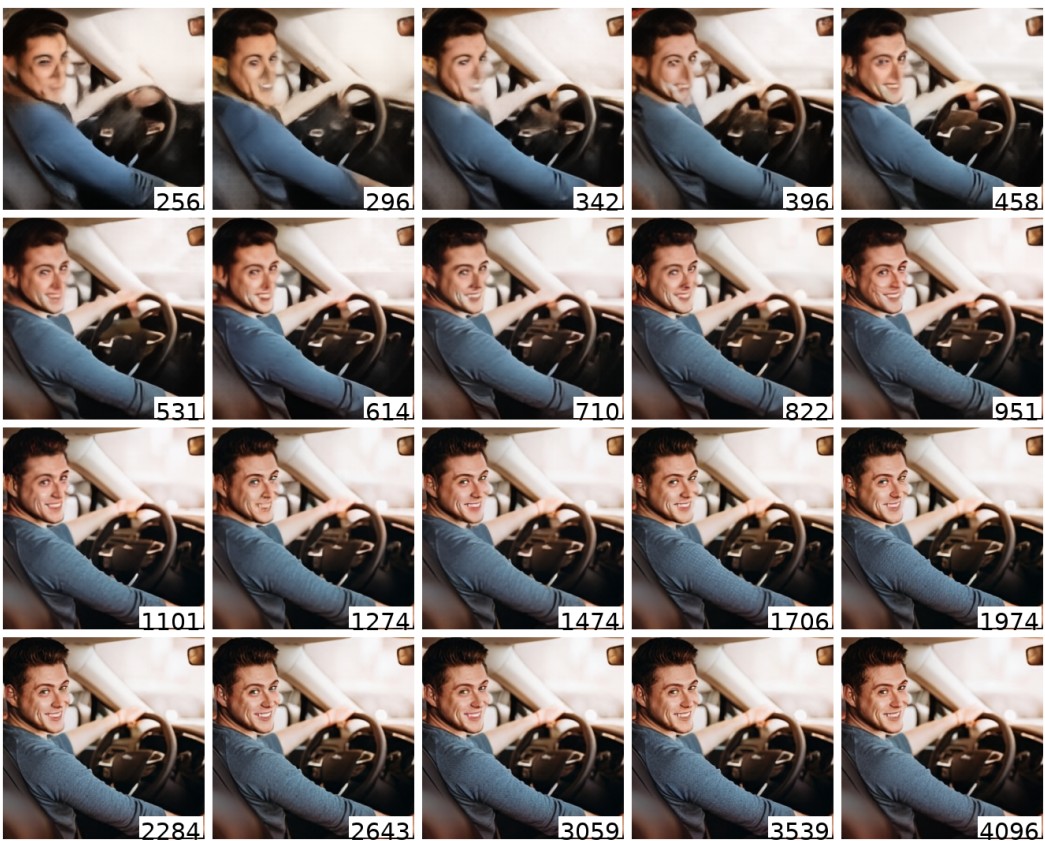

**Figure 11**  Progressive reconstruction of a given images as you increasing the number of tokens (left-to-right in the encoding). Images are in raster scan order, and the number of tokens used is in the bottom right of each image (from 256 - 4096 interpolated with 20 values on a log scale)

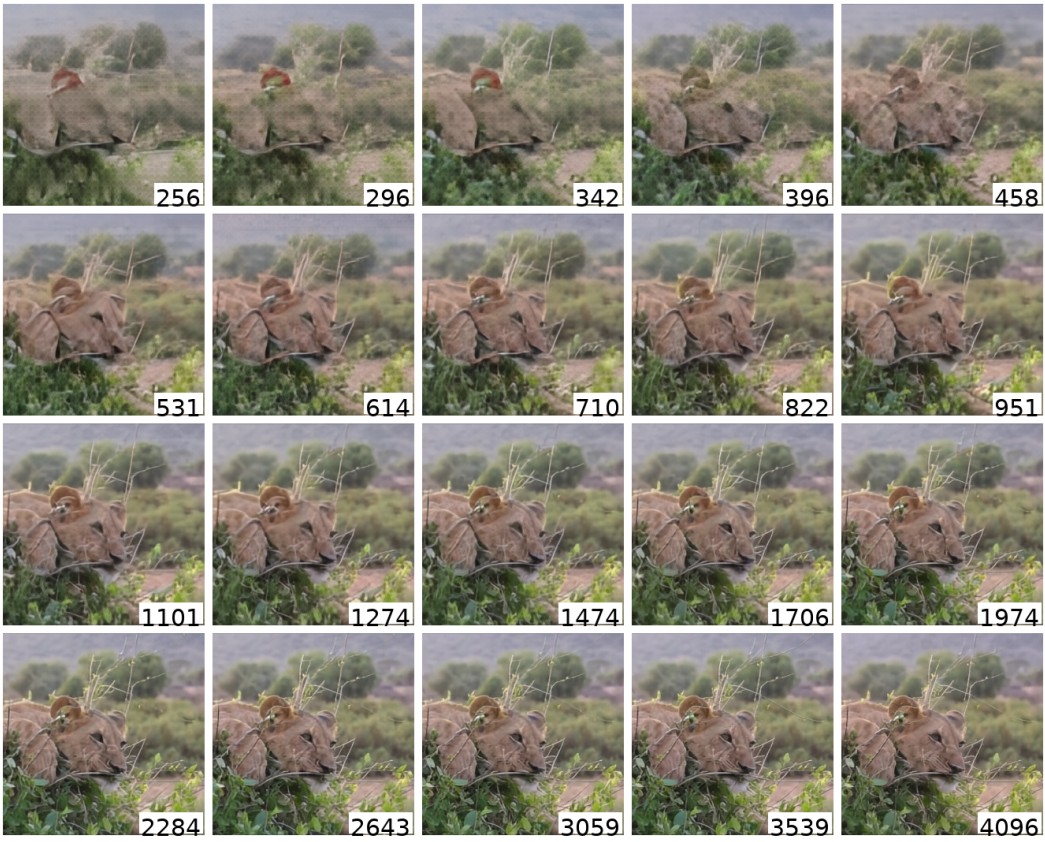

**Figure 12** Progressive reconstruction of a given images as you increasing the number of tokens (left-to-right in the encoding). Images are in raster scan order, and the number of tokens used is in the bottom right of each image (from 256 - 4096 interpolated with 20 values on a log scale)

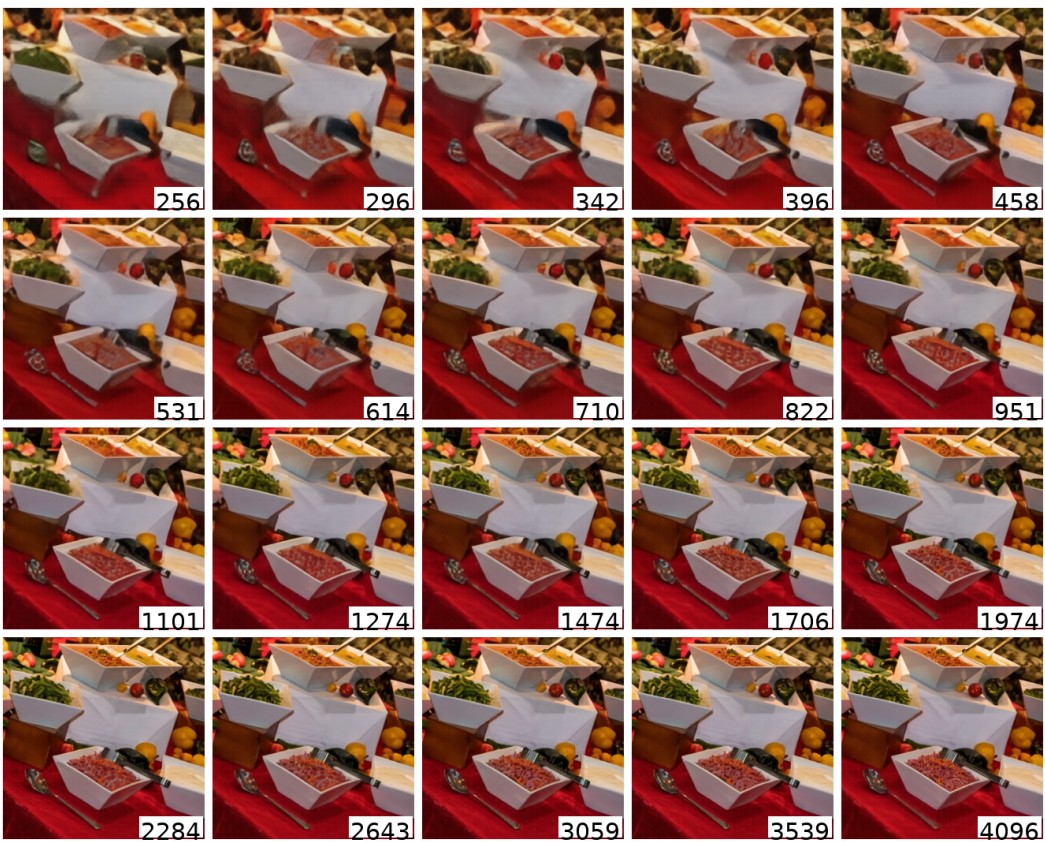

**Figure 13** Progressive reconstruction of a given images as you increasing the number of tokens (left-to-right in the encoding). Images are in raster scan order, and the number of tokens used is in the bottom right of each image (from 256 - 4096 interpolated with 20 values on a log scale)

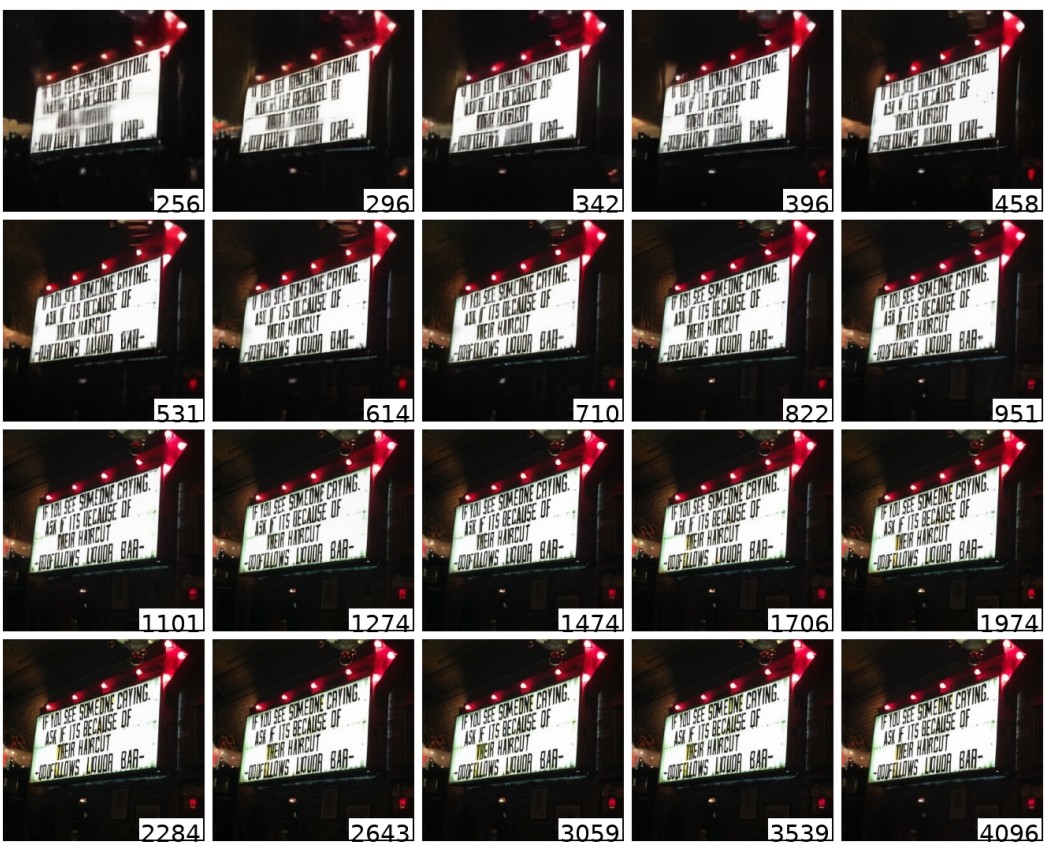

**Figure 14** Progressive reconstruction of a given images as you increasing the number of tokens (left-to-right in the encoding). Images are in raster scan order, and the number of tokens used is in the bottom right of each image (from 256 - 4096 interpolated with 20 values on a log scale)

