# OpenReview forum: "ElasticTok: Adaptive Tokenization for Image and Video"
_ICLR.cc/2025/Conference — ICLR 2025 Poster_

### Official Review · Reviewer_Qyem · 2024-11-01

**Soundness:** 3
**Presentation:** 3
**Contribution:** 2
**Rating:** 6
**Confidence:** 3

**Summary:**

This paper presents a method to encode images or videos with variable length tokens which can then be decoded in a manner that minimizes a reconstruction loss (or any other target loss). This is achieved by dropping (or masking) some portion of tokens during training and inference. The method assumes a fixed maximum token length l and chooses to use the first k (k<=l) tokens for encoding purposes. In case of longer videos, a block processing mechanism is adapted where a different number of tokens are used for each block.

**Strengths:**

- The idea of using adaptive token lengths for image/video encoding (or other sequential data) is valid especially since there is a desire to generate/analyze longer sequences of data.
- The paper is well written, it's easy to follow. The authors have also shared code.
- Provided experiments are reasonable with each experiment evaluating a clear aspect of the proposed approach.

**Weaknesses:**

- For me the biggest limitation of the method seems like although the paper claims to "adaptively" select the number of tokens given the input, it seems the current proposed strategies (e.g., binary search, trying to match a threshold etc.) still require various computation and hence might limit the applicability in certain scenarios. For example, adaptive token lengths might potentially be most useful in long generation tasks but it's not clear yet how one can adaptively decide how many tokens to generate.
- As the authors also mention in their limitations, the current masking strategy is pretty naive in the sense that it always uses the first k tokens. This may or may not be ideal depending on the input content.

**Questions:**

For Fig.4 comparisons, do the authors first search for the ideal token length for each input for their method?

In Table3, if I understand correctly the baseline is trained with 50% of tokens and the method is trained with varying number of tokens. How is the inference done? Do both methods use 50% of tokens during inference?

---

> ### Author Response · Authors · 2024-11-18
> **Rebuttal**
>
> Thank you for your constructive feedback. We provide detailed answers to the questions below.
>
>
> ---
> > **Choosing how to adaptively encode, or generate tokens**
>
> As mentioned by the reviewer, a primary method of adaptively selecting the number of tokens is encoding image or video according to a pre-defined threshold. Downstream generative models can still generate a variable number of tokens by conditioning on a threshold, or having no conditioning but training on a fixed threshold. For example, generating a video at a target MSE of 0.003 will have a model dynamically generate a different number of tokens per video block depending on the underlying dynamics and complexity of the generated video. This would require running ElasticTok inference at 0.003 threshold, and training on the resulting encodings.
>
> ---
> > **First-k tokens masking limitation**
>
> Although the left-aligned masks are simple, we believe that our experiments (e.g. Fig 4) show the effectiveness of our method on general image and video. Both training and evaluation distributions span a diverse range of visual content from the web – cartoons, realistic movies, fast transitions, scene cuts, etc. The current approach strikes a practical balance between computational overhead and effectiveness. However, we do agree that better masking schemes (e.g., content-adaptive masks or attention-guided token selection) could further boost performance, which we plan to explore in future work.
>
> ---
> > **For Fig.4 comparisons, do the authors first search for the ideal token length for each input for their method?**
>
> Yes, each image / video is adaptively encoded to achieve the target MSE. This allows the model to potentially allocate more tokens to harder to encode images while still retaining lower average token usage. In contrast, baseline models must always encode with the same number of tokens, which may waste tokens on easy to encode images, or be too lossy on difficult images.
>
> ---
> > **Do both methods use 50% of tokens during inference in Table 3 experiments?**
>
> The reviewer asked for clarification about inference token usage for both methods in Table 3.
> The baseline is trained with 100% of tokens, and ElasticTok is trained with a varying number of tokens. Inference in Table 3 is comparing the baseline and ElasticTok both using 100% of tokens during inference, demonstrating that ElasticTok can match baseline performance when using full context, while maintaining the additional advantage of allowing users to choose token-accuracy tradeoffs during inference if desired (e.g. able to perform more API queries if choosing to use fewer tokens).

---

> > ### Comment · Reviewer_Qyem · 2024-11-26
> >
> > Thank you for the reply. Based on your answers also to reviewer bdcm, I think I now have a better understanding of the system. It was not indeed clear to me that the encoder would also take a mask as input. I think this part should be made clear in the final version. I also understand the proposed strategy of how to train a video generation model with Elastic Tok. How would we adopt this for inference time in that case though? Would a generator still generate a fixed number of tokens or would one adapt like an autoregressive approach?
> >
> > While I still have questions around how this approach could be applied to practical use cases, overall I'm satisfied with the rebuttal of the authors and remain positive.

---

> > > ### Author Response · Authors · 2024-11-27
> > > **Response to Reviewer Qyem**
> > >
> > > We would like to thank the reviewer for their positive reply. We apologize for the confusion with regards to the encoder masking, and have updated the Fig 2 in the paper to more clearly show this.
> > >
> > > Regarding the technical question on applying ElasticTok to downstream video generation models: the generator would generate a variable number of tokens along with special tokens. During training, one can pack variable-length sequences together into fixed length sequences to train on variable-length generation tasks. Importantly, also include special EOS_1, EOS_2 tokens to represent the end of a single video block, and then end of the entire video respectively. Then, two potential modelling options are:
> > > * Train an autoregressive model. In this case the AR model (LLM-like with discrete, or ARDiffusion [1] for continuous encodings) would just generate until it decides to terminate by generating EOS_2.
> > > * Train a bidirectional model such as a diffusion transformer [2] or MaskGit [3] on variable length sequences, along with a categorical prior on the distribution of encoding lengths for each block in the video. The sampling process would be to (1) sample the token lengths, then (2) sample the whole video with a noise sequence constructed for the token lengths (with unnoised EOS_1, EOS_2 tokens).
> > >
> > > For both cases, decoding would separate the result into individual video blocks by slicing on EOS_1 tokens, and then feeding each block causally through the ElasticTok decoder to produce RGB frames.
> > >
> > > This natural integration of ElasticTok with training practical models would allow ElasticTok's variable-length encoding to naturally handle real-world scenarios in an efficient manner, where video content has different levels of complexity and motion.
> > >
> > > We would be happy to further address any additional specific questions and concerns you may have!
> > >
> > >
> > > [1] Li, Tianhong, et al. "Autoregressive Image Generation without Vector Quantization." arXiv preprint arXiv:2406.11838 (2024).
> > >
> > >
> > > [2] Peebles, W., & Xie, S. "Scalable diffusion models with transformers." Proceedings of the IEEE/CVF International Conference on Computer Vision. 2023
> > >
> > >
> > > [3] Chang, Huiwen, et al. "Maskgit: Masked generative image transformer." Proceedings of the IEEE/CVF Conference on Computer Vision and Pattern Recognition. 2022.

---

### Official Review · Reviewer_xRJN · 2024-11-03

**Soundness:** 2
**Presentation:** 3
**Contribution:** 2
**Rating:** 6
**Confidence:** 3

**Summary:**

This paper proposes a video representation learning approach that learns a compressed representation via an autoencoder. Randomly dropping the last few tokens of each frame during training encourages important information to stay in the first few tokens. This allows users to dynamically adjust the number of tokens used during inference, balancing the trade-off between faster inference and higher output quality based on the task's requirements.

**Strengths:**

Exploiting temporal coherence to learn a more compact video representation is a well-motivated problem.
Figure 4 does indeed show that the dropout strategy can reduce the number of tokens as intended, although it requires a computationally exhaustive search to find the right number of tokens for each frame.

**Weaknesses:**

The "neural regression" evaluation (Table 4) does not seem very helpful. If I understand correctly, the error rate is the rate the regressor failed to correctly predict the optimal number of tokens found by the exhaustive search. It would have been good to also show the reconstruction accuracy. Also it would have been good to show the total inference time or FLOPS for each search method. It can be architecture-dependent but how much computation can be saved using fewer tokens in the decoder should be more carefully presented considering claims about efficient computation.

**Questions:**

I think it might be helpful to mention which search method was used for each evaluation (e.g. in Figure 4). I was assuming exhaustive search was used for most if not all of them.

I could not find details on how the regressor is fine-tuned. Does it see the context of the entire video? Does the regression loss change the encoder weights?

Experiment suggestion: a heuristic-based regressor baseline (e.g. use the amount of motion as a proxy to determine the number of tokens needed)

Citation suggestion: "Adaptive Computation Time for Recurrent Neural Networks" by Alex Graves

---

> ### Author Response · Authors · 2024-11-18
> **Rebuttal**
>
> Thank you for your constructive feedback. We provide detailed answers to the questions below.
>
> ---
> > **Total inference time of each method**
>
> Table 4 shows the inference time in terms of NFEs, or the number of forward passes needed for the model, and the exact model used. However, the total FLOPs can be directly computed as the model FLOPs for a single forward pass times the NFEs. For example, the FLOPs for binary search on the continuous ElasticTok model is approximately (819 GFLOPs * 12 NFEs = 9.8 TFLOPs).
>
>
> ---
> > **Claims on efficient computation**
>
> We would like to clarify that the tokenizer itself does not save computation, as the encoder and decoder always operate on the full sequence length (the decoder just sees a masked / padded input). **However**, our experiments show that adaptive encoding (Fig 4) results in large savings on **token lengths of resulting encoding sizes**, which will save 2-5x tokens for downstream pretrained VLMs. This **directly translates to** at least 2-5x (more as sequence length grows due to quadratic attention) speedup in training and inference. This is especially valuable as pretrained downstream models are very large (10Bs - 100Bs params with at least ~2-5x speedup), and the extra computation from the tokenizer (200M params) is less significant and can be pre-encoded.
>
>
> ---
> > **Search method used for each evaluation**
>
> Thank you for this clarification. Unless otherwise noted, we use **binary search** as the search algorithm for all results. We have edited the paper (L239) to accurately reflect this.
>
>
> ---
> > **Details on the regressor**
>
> We have added more details in Appendix E on how the regressor is finetuned. The encoder is frozen. A small number of extra trainable transformer layers are added, along with progressive downsampling to eventually output a single scalar, which is the predicted number of tokens. The targets are normalized to [-1, 1], and regress to using MSE. The regression model is also block causal, and does see the entire historical context of the video.
>
> Motion-representations would be an accurate source of information with respect to slow / fast changes in the video, however do not generalize well to all cases, such as abrupt scene cuts (very common in videos), and frames which are just dense in visual information even if there is little motion (a very cluttered room, or a video of just static noise). Hence, we decided to use pixel representations of videos in our regression models as it contains complete information about any relevant video statistics.
>
> ---
> > **Citation suggestion**
>
> Thank you for this reference. We have added it as further discussion in our related works (L472-473).

---

> ### Author Response · Authors · 2024-12-02
> **Followup on Rebuttal**
>
> Dear Reviewer xRJN,
>
> Thank you for taking the time to review our paper. We appreciate your feedback and believe we have addressed the reviewer’s concerns regarding efficiency, search methods, and regressor details in our rebuttal. We would greatly appreciate your consideration of our response.
>
> Best regards,
>
> Authors

---

> > ### Comment · Reviewer_xRJN · 2024-12-03
> >
> > Thanks for the reply. After reviewing other reviews and the responses, now I understand the approach better now. My concerns have been addressed. I would like to change my rating to above the threshold. Thank you

---

> > > ### Author Response · Authors · 2024-12-03
> > > **Official Comment by Authors**
> > >
> > > Dear Reviewer xRJN,
> > >
> > > Thank you for your positive evaluation and for raising your rating. We appreciate your thorough review of our work and the time you took to consider our rebuttal.
> > >
> > > Best,
> > >
> > > Authors

---

### Official Review · Reviewer_xU6C · 2024-11-04

**Soundness:** 3
**Presentation:** 3
**Contribution:** 2
**Rating:** 6
**Confidence:** 3

**Summary:**

This paper proposes ElasticTok, a method for adaptive image/video tokenization that encodes visual data into variable-length tokens.  During training, the method drops a random number of tokens at the end of the token encoding. During inference, ElasticTok can use a variable length of tokens according to the need. Experiments show the advantages of the proposed method compared with the baseline.

**Strengths:**

1. This paper proposes an effective way to compress the number of tokens adaptively.

2. The length of used tokens aligns well with human intuition: when the image is more complex, more tokens are used.

3. The proposed method can effectively represent long videos with up to 2-5x fewer tokens.

**Weaknesses:**

1. Lack of comparison. The proposed method should be compared with other state-of-the-art token compression methods considering the compress rate, reconstruction quality, and downstream task performance.

2. Loss of details in reconstruction. From the demos, the reconstruction quality is not very satisfactory. For example, the loading icon on the cellphone has totally disappeared/distorted.

3. More analysis of the drop in image quality when using the elastic token compression is desired. Consider measuring the quality drop between the original image and the reconstructed image using PSNR/SSIM.

**Questions:**

See Weakness.

Besides, the applications of elastic token compression are still not so convincing to me. Are there more applications that clearly benefit from using elastic token compression than from using other compression methods?

---

> ### Author Response · Authors · 2024-11-18
> **Rebuttal**
>
> Thank you for your constructive feedback. We provide detailed answers to the questions below.
>
> ---
> > **Lack of comparisons**
>
> Our paper focuses on developing an autoencoder-based adaptive tokenizer that can be used for downstream any-to-any multi-modal models (e.g. Chameleon [1], Omnigen [2]). Thus, we focus on comparing against SoTA baseline tokenizers that these models use – generally consisting of fixed-length autoencoders that learn discrete (VQ, FSQ), or continuous (VAE) latents using either CNNs or transformers.
>
> Since our only difference with prior SoTA tokenizers is solely fixed vs elastic (variable) length encodings, we design baseline comparison experiments that retain aspects of standard tokenizer architectures (ViT architecture, MSE + LPIPs losses) and only alter the model to learn elastic encodings. We believe that this is the most fair and direct comparison with SoTA tokenization methods, and our comparable token usage efficiency demonstrates the benefits of our method.
>
> [1] Meta AI. "Chameleon: Mixed-modal early-fusion foundation models." arXiv preprint arXiv:2405.09818 (2024).
>
> [2] Xiao, S., Wang, Y., Zhou, J., Yuan, H., Xing, X., Yan, R., ... & Liu, Z. (2024). Omnigen: Unified image generation. arXiv preprint arXiv:2409.11340.
>
>
> ---
> > **Reconstruction quality when using elastic token compression**
>
> Fig 4 shows the reconstruction quality performance of our model – ElasticTok achieves better reconstruction of images/videos (by MSE/PSNR) compared to the baseline while using fewer percentages of tokens. Note in general some reconstructions may be a little blurry since we primarily used MSE as the main optimization loss. Our proposed method is orthogonal to additional perceptual objectives such as GAN losses, which have shown to drastically improve image sharpness and can be additionally applied.
>
>
> ---
> > **Applications of elastic token compression, and compared to other compression methods**
>
> To the best of our knowledge, all existing any-to-any multi-modal models use fixed length tokenizers. This is generally undesirable, as allocating a fixed number of tokens for all images / videos will result in wasted tokens for simple images, and lossy encodings for complex images. With ElasticTok, we believe that a paradigm shift towards elastic (variable) token lengths can more efficiently allocate computational costs for downstream training and inference, which is generally by far the more expensive part in current generative models. Our model demonstrates up to 2-5x savings (Figure 4) in encodings lengths for images and long videos, which directly translates to at least 2-5x savings in training and inference costs of current generative models that traditionally used fixed length tokenizers.

---

> > ### Comment · Reviewer_xU6C · 2024-11-27
> >
> > Thank the authors for the reply. After carefully reviewing the rebuttal and discussions with other reviewers, I am pleased to note that most of my concerns have been addressed. Based on this, I am inclined to raise my rating.

---

> > > ### Author Response · Authors · 2024-11-30
> > >
> > > Dear Reviewer xU6C,
> > >
> > > Thank you for your positive evaluation and for raising your rating. We appreciate your thorough review of our work and the time you took to consider our rebuttal.
> > >
> > > Best,
> > >
> > > Authors

---

### Official Review · Reviewer_bdcm · 2024-11-04

**Soundness:** 3
**Presentation:** 1
**Contribution:** 3
**Rating:** 6
**Confidence:** 5

**Summary:**

- The paper presents an approach to dynamically assign different number of tokens per different image/video depending on the input complexity.
- During training, a random mask is applied to the encoded image tokens, and the unmasked tokens are used to reconstruct the image. - For training the video pipeline, similar random masking is applied at per block (stack of 4 images) levels, with the masking function being independent per block.
- During inference, an iterative search is performed over a different number of tokens to be masked, and the max mask that satisfies the required constraint (reconstruction loss < target reconstruction loss) is selected.
- Interesting results – analyzing the performance of image/video reconstruction quality and fine-tuned VLM performance at different token counts.

**Strengths:**

- The paper studies an interesting topic of adaptive tokenization for images and videos. As mentioned, such adaptive tokenization is crucial for long-context tasks.
- While similar adaptive and compression architectures have been studied in the past, the application to auto-encoding of images and videos is a novel contribution. Related works in adaptive representations and compression are well covered.
- Results include interesting analysis of token requirement for reconstruction and VLM tasks.

**Weaknesses:**

- The paper writing needs improvement –
    - **A background section on Ring attention** (Liu et al., 2024b) paper should be the first paragraph of the method section since Ring attention is a big component of the proposed approach. Most important are the **details of the autoregressive video model**.
    - Algorithm 1 and 2 says variable "x" (output of PatchifyRearrange) shape is B $\times$ L $\times$ D, what exactly is L ? From the next line ($N_b$ = the number of blocks), it seems that "x" should be of shape: B $\times$ ($\frac{L}{B}$) $\times$ D i.e. number of tokens per block $\times$ number of blocks $\times$ feature dim. Please clarify this and **define all variables**.
    - The captions of Tables 1 and 2 and the corresponding paragraph writing can be improved. It seems that table 1 is for image dataset and table 2 is for video dataset, but this is not mentioned anywhere. Please clarify and **fix these writing issues**.

- Questions regarding masking strategy: Encoders / Tokenizers like VAE and VQGAN map tokens to patches and also assign 2D positional encodings to each patch. Thus, in some sense, each token (although captures enough receptive field via convolution or self-attention) is responsible for the reconstruction of the corresponding patch. Architectures like MAE mask random patches/tokens, performing a local task of inpainting (because at least some token is unmasked in the vicinity of a masked token). But the proposed approach always masks the leftmost tokens meaning that a fixed section of the image is always masked, irrespective of the content complexity of that particular section of the image. This raises questions on the choice of masking and doubts about the efficacy of the trained model. Visualizing what the leftmost tokens capture could help understand the issue. Details about position encodings would also be great.

- Does the encoder take the sampled mask as input? While the text says so, the architecture figure shows that the mask is applied after full-length encoding.

- For image reconstruction, searching for the mask at inference time makes sense, but for video level, searching for masks seems too computationally expensive. The only possible approach seems neural regressive in that case. More details on the neural regression approach would be great, including details on whether the neural masking approach is independent per frame, or it takes into account the context like the autoregressive video encoder.

- For video training, independent per-block mask sampling will lead to a combinatorially large number of masking choices per video, with a majority of them being not very useful for training. For example, strong masking of the first frame, and no masking of the immediate next frame might be a waste if there is no motion from one frame to another. Thus, a next-frame masking strategy should also have better inductive biases in order to train larger models efficiently.
- More details about the video dataset will be great. Will the exact video dataset be released to support the reproduction of the proposed approach?
- It seems that all paper results are from the continuous model. Results or ablations from the quantized FSQ model would be appreciated.

**Questions:**

My main concern are weakness points highlighting lack of clarity or details over the left-most token masking strategy. This raises concerns regarding efficacy of the trained model. But I feel the authors should be able to address this easily both in text and by visualizing the tokens.

The second concern is that independent per frame or per block masking strategy doesn't seem to be a scalable or efficient training approach. Intuitive understanding of the authors and maybe some statistics of the masking done during training could help alleviate these issues.

Rest, writing should be improved and this should be also fixable.

---

> ### Author Response · Authors · 2024-11-18
> **Rebuttal (1/2)**
>
> Thank you for your constructive feedback. We provide detailed answers to the questions below.
>
> ---
> > **Paper presentation suggestions**
>
> We thank the reviewer for their thoughtful suggestions. We have addressed all points in our revised manuscript (changes highlighted in magenta):
> * Added RingAttention background: We have added a comprehensive background to explain RingAttention's core mechanisms and how these enable efficient processing of long sequences critical to our approach (Section2, L79-115)
> * Added more architectural details on the video model (Section 4.1, L215-220)
> * Added more details on the variables, and shapes in the algorithms with clear variable definitions, shape information, and step-by-step explanations (Algorithm 1 and 2, L187-204)
> * Clarified Tables 1 and 2 with more detailed discussion for each (Moved to Appendix F due to space constraints from prior changes, L850-886)
>
>
> ---
> > **Does the encoder take the masked samples as input?**
>
> Yes, the encoder does take the masked samples as input. We appreciate your careful attention to this technical detail. We have updated Figure 2 to explicitly show the masked samples flowing into the encoder block, making the architecture diagram more accurate and clearer.
>
>
> ---
> > **Masking strategy and positional embeddings**
>
> We use 1D positional embeddings over the entire sequence length, treating a patchified video as a flattened sequence of patches in raster scan order. We empirically found this simple approach effective at capturing spatiotemporal relationships, as evidenced by our model's strong performance across temporal tasks.
>
> Importantly, we **do not** left-mask the input video itself, but rather apply masking only to the resulting encoding. Thus, the encoder is given the whole unmasked video as well as the mask that **will be** applied so it can learn to put as much information as it can into the left-most tokens.
>
> This allows our encoder to process the complete unmasked video while being aware of which tokens will be masked, enabling it to optimize the information packing into the left-most tokens that will be preserved.
>
> The nature of the representations learned is hierarchical in a coarse-to-fine manner. This can be easiest seen in the progressive reconstructions (e.g. Figure 5, where we show video reconstructions using an increasing number of tokens from left to right, starting with the minimum token count).
>
> To further illustrate this coast-to-fine manner, we have added more detailed progressive reconstruction examples in Appendix G, along with specific token counts for each reconstruction stage in the bottom right of each image.
>
> ---
> > **Inference Time Compute Cost**
>
> Searching for masks at inference time indeed requires multiple function evaluations at video level, we would like to highlight a few points regarding how this overhead is both manageable and justified:
>
> * **Tokenizer vs Pretrained Model sizes**: Tokenizers are fairly lightweight compared to downstream vision-language models (200-500M tokenizers vs 10B-100B downstream VLMs). Thus, extra computation time of the tokenizer (1-10x depending on usable search algorithms) is less significant compared to running the downstream model. Additionally, the extra compute differences can also be made up through fewer flops due to token length reduction for the downstream model.
> * **Parallelizability**: Binned search methods such as 100 bins (0.5% error) or 10 bins (5% error), can be parallelized by just increasing batch size (compute different masking levels in parallel). In the best case, we can leverage more compute to have inference equivalent in speed to 1 forward pass / baseline models.
> * **Future work in optimized implementations**: While it is out of scope of this work, our inference method is similar to language models (block causal instead of standard causal), and can leverage many optimization techniques to speed up inference, such as quantization, paged attention / more efficient caching, more efficient kernels, etc.
>
> Overall, we do not believe that inference speed is a major bottleneck, and the benefits of adaptive tokenization outweigh the potential (addressable) drawbacks – ElasticTok enables more efficient token usage in downstream tasks, of which models are generally much larger, and more compute intensive.

---

> ### Author Response · Authors · 2024-11-18
> **Rebuttal (2/2)**
>
> ---
> > **Combinatorial masking combinations on long sequences**
>
> We appreciate  the reviewer’s suggestions on making our combinatorial masking on long sequences more scalable and efficient.
>
> Our masking is indeed combinatorial, but it scales well to longer sequences. Our training on up to 1M tokens length sequences, the training is stable and works reasonably well on long sequences during inference. Figure 6 shows evidence of this, with monotonically increasing   encoding performance on longer sequences up to 512K tokens, except the slight performance drop at 1M tokens likely due to insufficient gradient updates rather than a fundamental limitation.
>
> Intuitively, we are training on every masking task, which may help the model generalize to a wider range of unknown inference tasks, such as a more lossy encodings (higher masking amounts), less lossy encodings (lower masking amounts), or a mix (e.g. inference loss functions that only prioritize higher reconstruction quality on human faces, while lossily encoding everything else).
>
> We do agree that better masking schedules could improve the training efficiency of our model when seeking to apply our model for specific use case, perhaps through some heuristic method, or potentially bootstrapping off our current model. We believe that our current simple masking method demonstrates strong empirical results while maintaining implementation simplicity and represents a valuable contribution to the field, while leaving the door open for future optimizations.
>
>
> ---
> > **Video datasets release**
>
> Thank you for this question. Additional details about the video dataset construction, including data sources, sampling methodology, and preprocessing pipeline, are provided in Appendix C. We have eventual plans to publicly release the video dataset to support reproducibility, however, it is still in the process of being further filtered to exclude unwanted (e.g. NSFW) data (~1% of the current dataset) before release.
>
>
> ---
> > **Results (discrete vs continuous)**
>
> Both discrete and continuous models generally perform similarly, and experimental results show a mix of discrete and continuous models. We apologize for any confusion about model types in our results. The core results presented in Figure 4 were actually from the FSQ model, not the continuous model. We have now clarified this in the figure caption, and updated other table / figure captions in the paper. For comparison, Appendix F Table 3 presents parallel results from the continuous VAE model, demonstrating consistent performance across both architectures.

---

> ### Comment · Reviewer_bdcm · 2024-11-24
> **Question Regarding Masking Strategy and Baseline**
>
> Firstly, I appreciate the authors for putting up the rebuttal.
>
> > Importantly, we do not left-mask the input video itself, but rather apply masking only to the resulting encoding. Thus, the encoder is given the whole unmasked video as well as the mask that will be applied so it can learn to put as much information as it can into the left-most tokens.
>
> \
> Initially, my understanding—due to the lack of masking details in the original submission—was that the encoder learns fixed-length embeddings, which are later masked. Therefore, as the authors mentioned in the related work, ElasticTok appeared to be an application of the Matryoshka approach to image/video reconstruction.
>
> However, based on the authors’ latest response, it seems the encoder is run repeatedly for different sampled masks at test time. This approach appears highly inefficient and results in wasted computational flops (doesn't seem a scalable approach for video going forward, if for each frame the whole encoder needs to be run again and again to find optimal mask).
>
> Why not adopt a single embedding, as in the Matryoshka approach, and perform masking directly on the encoded output? This would serve as an important baseline for comparison, and I would appreciate it if the authors could provide results for this approach.

---

> > ### Author Response · Authors · 2024-11-26
> > **Response to Reviewer bdcm**
> >
> > Thank you for the response, and apologize for any confusion caused in our figures. We initially adopted a single embedding as in the Matryoshka approach, but chose to additionally condition the encoder as we saw slight benefits in doing so. We have added an additional ablation (Appendix F, Table 5) and additional discussion about the associated inference cost (L183-185) in the paper. The ablation compares running ElasticTok with and without encoder conditioning – included below for convenience.
> >
> > |                  |  0.001 |  0.003 |  0.015 |
> > |------------------|:------:|:------:|:------:|
> > | With Enc Cond    |  **36.2\%**  | **79.6\%** | **98.5\%** |
> > | Without Enc Cond | 33.1\% | 75.3\% | 98.4\% |
> >
> > The table shows the percentage of auto encoded video blocks that satisfy each desired reconstruction threshold (0.001, 0.003, 0.15), assuming that both models are constrained to use the same average tokens used. In general, encoder conditioning provides only modest improvements in performance. For large downstream models (20B - 100B+), these small improvements become more valuable: a small reduction in encoded token lengths can reduce a significant amount of FLOPs, outweighing the doubled inference cost of the tokenizer itself. However, for smaller downstream models, the trade-off favors the simpler approach: using Matryoshka-like encodings without encoder conditioning maintains most of the benefits of elastic encodings while halving the inference cost.
> >
> > We hope that this addresses the reviewer’s concerns, and would be happy to further discuss any additional concerns.

---

### Meta-Review · Area_Chair_MNuz · 2024-12-14

**Metareview:**

The paper proposes an adaptive approach to encoding frames into tokens to substantially reduce the number of tokens to be processed in downstream tasks.

Reviewers appreciated the novel application of adaptive tokenization to auto-encoding images and videos. They considered the token reduction effective and contributions sufficiently validated through experiments.

Initially, reviewers were concerned about the effectiveness of the encoding due to the need of a search for the right number of tokens at inference. This, and further concerns about the masking strategy, image quality, and implementation details were clarified during the discussion phase. Although all concerns were clarified and reviewers raised their scores correspondingly, no reviewer championed the paper and the overall score remains marginally above acceptance.

**Additional Comments On Reviewer Discussion:**

- exhaustive search needed for efficient encoding, neural regressor inaccurate [xRJN], clarified in discussion (default binary search)
- missing comparisons to relevant baselines [xU6C], clarified in discussion
- insufficient analysis of image quality [xU6C], clarified in discussion
- potentially suboptimal masking strategy [bdcm], clarified in discussion & revision
- missing details on neural regression, inefficiency for video [bdcm]
- missing background on ring attention [bdcm], addressed in revision
- missing details on architecture [bdcm], addressed in revision
- inefficient training strategy for video [bdcm], clarified in discussion

---

### Decision · Program_Chairs · 2025-01-22

Accept (Poster)